# BATCH NORMALIZATION HAS MULTIPLE BENEFITS: AN EMPIRICAL STUDY ON RESIDUAL NETWORKS

## ABSTRACT

Many state of the art models rely on two architectural innovations; skip connections and batch normalization. However batch normalization has a number of limitations. It breaks the independence between training examples within a batch, performs poorly when the batch size is too small, and significantly increases the cost of computing a parameter update in some models. This work identifies two practical benefits of batch normalization. First, it improves the final test accuracy. Second, it enables efficient training with larger batches and larger learning rates. However we demonstrate that the increase in the largest stable learning rate does not explain why the final test accuracy is increased under a finite epoch budget. Furthermore, we show that the gap in test accuracy between residual networks with and without batch normalization can be dramatically reduced by improving the initialization scheme. We introduce "ZeroInit", which trains a 1000 layer deep Wide-ResNet without normalization to 94.3% test accuracy on CIFAR-10 in 200 epochs at batch size 64. This initialization scheme outperforms batch normalization when the batch size is very small, and is competitive with batch normalization for batch sizes that are not too large. We also show that ZeroInit matches the validation accuracy of batch normalization when training ResNet-50-V2 on ImageNet at batch size 1024.

## 1 INTRODUCTION

Skip connections (He et al., 2016a) and batch normalization (Ioffe & Szegedy, 2015) have contributed to the dramatic improvement in the performance of convolutional nets in recent years. However batch normalization also has a number of limitations. It breaks the independence between training examples within a batch, performs poorly with small batch sizes (Wu & He, 2018), and increases the cost of computing an update (Wu et al., 2018). Many authors have tried to understand the role batch normalization plays during training (Balduzzi et al., 2017; Santurkar et al., 2018; Bjorck et al., 2018; Sankararaman et al., 2019; Yang et al., 2019; Ghorbani et al., 2019), to replace batch normalization with cheaper alternatives (Ba et al., 2016; Salimans & Kingma, 2016; Ulyanov et al., 2016), or to remove normalization entirely (Mishkin & Matas, 2015; Krähenbühl et al., 2015; Zhang et al., 2019).

Santurkar et al. (2018) argued that batch normalization helps primarily because it improves the conditioning of the loss landscape, enabling stable training with larger learning rates. Bjorck et al. (2018) argued that this not only explains why batch normalization helps during optimization, but also explains why it improves the final test accuracy, since large learning rates are thought to generalize well (Keskar et al., 2016; Smith & Le, 2017; Jastrzębski et al., 2017). Other authors have remarked that batch normalization introduces noise by computing the batch statistics over a subset of the training data, helping generalization (Luo et al., 2019). Meanwhile, both Balduzzi et al. (2017) and Yang et al. (2019) argued that the combination of batch normalization and skip connections preserves the correlation between the gradients of different training examples at initialization, and they speculated that this enables the efficient training of deep architectures. Mishkin & Matas (2015), Krähenbühl et al. (2015) and Zhang et al. (2019) have all demonstrated empirically that some of the benefits of batch normalization can be recovered by improving the initialization scheme.

In this work, we argue that one cannot build a complete picture of the benefits of batch normalization, unless one studies the training loss and test accuracy achieved by deep networks at a *wide range of batch sizes*. When the batch size is large, one can train with large learning rates, and the role of batch normalization in enhancing the conditioning of the loss landscape becomes apparent. When the batch

size is smaller, one does not need large learning rates to train efficiently, and so we can study the other benefits of batch normalization in isolation. Our results establish the following:

1. Batch normalization does enable training with significantly larger learning rates, and this enables practitioners to efficiently parallelize training over much larger minibatches.

2. However this is not the only benefit. Batch normalization also performs well on both the training loss and the test accuracy with small batch sizes and small learning rates.

3. We propose "ZeroInit", a simple change to the initialization scheme of residual networks, designed to preserve the correlation between gradients at the start of training. We show that this enables us to train very deep residual networks without normalization, and our method recovers most of the benefits of batch normalization when the batch size is not too large.

Our results suggest that batch normalization improves conditioning (Santurkar et al., 2018; Bjorck et al., 2018), preserves the correlation between gradients (Balduzzi et al., 2017; Sankararaman et al., 2019; Yang et al., 2019), and has additional regularization benefits (Luo et al., 2019). It is easy to miss one or more of these benefits if one performs experiments at a single batch size. One should also be cautious in claiming to identify a complete replacement for batch normalization without first verifying that it matches the performance of batch normalization for a wide range of batch sizes.

This paper is structured as follows. In section 2 we introduce residual networks, batch normalization, and Fixup initialization (Zhang et al., 2019). In section 3 we compare the test accuracy and training loss of residual networks trained with and without batch normalization at a range of batch sizes, with the aim of isolating the different benefits batch normalization brings. In section 4 we introduce "ZeroInit", a simple initialization scheme inspired by both Fixup and shattered gradients (Balduzzi et al., 2017). This scheme trains very deep residual networks to high accuracies without batch normalization, but it does not replicate the full benefits of batch normalization when training with large batch sizes. In section 5 we evaluate the performance of ZeroInit and Fixup on ImageNet.

## 2 BACKGROUND

**Residual Networks (ResNets):** First introduced by He et al. (2016b), these models are composed of a sequence of residual blocks, which contain both a "residual branch" containing a number of convolutions and activation layers, as well as a "skip connection", which is typically just the identity. These skip connections create pathways from the input to the output which have a shorter effective depth than the main path through all the residual branches, and these pathways enable the input signal to pass more easily from the input to the output (Balduzzi et al., 2017; Sankararaman et al., 2019). Crucially, the network still retains a high degree of expressivity thanks to the main path through all the residual branches. Most of the experiments in this paper will follow the popular Wide-ResNet architecture, introduced by Zagoruyko & Komodakis (2016). However we also consider the original ResNet-V1 (He et al., 2016a) and ResNet-V2 (He et al., 2016b) architectures in section 5.

**Batch Normalization:** As in most previous work, we apply batch normalization in convolutional layers. The inputs to and outputs from batch normalization layers are therefore 4-dimensional tensors, which we denote by $I_{b,x,y,c}$ and $O_{b,x,y,c}$. Here $b$ denotes the minibatch, $c$ denotes the channels, and $x$ and $y$ denote the two spatial dimensions. Batch normalization applies the same normalization to every input in the same channel (Ioffe & Szegedy, 2015). This leads to the update:

$$O_{b,x,y,c} = \gamma_c \frac{I_{b,x,y,c} - \mu_c}{\sqrt{\sigma_c^2 + \epsilon}} + \beta_c.$$

Here, $\mu_c = \frac{1}{Z} \sum_{b,x,y} I_{b,x,y,c}$ denotes the per-channel mean, and $\sigma_c^2 = \frac{1}{Z} \sum_{b,x,y} I_{b,x,y,c}^2 - \mu_c^2$ denotes the per-channel variance of the inputs, and $Z$ denotes the normalization constant, which is summed over the minibatch and spatial dimensions $x$ and $y$. A small constant $\epsilon$ is generally added to the variance for numerical stability. The "scale" and "shift" parameters, $\gamma_c$ and $\beta_c$ respectively, are learnt during training and are added to preserve model expressivity after the normalization operation. Running averages of the mean $\mu_c$ and variance $\sigma_c^2$ are also maintained during training, and these averages are used at test time to ensure the predictions are independent of other examples in the batch.

Batch normalization enables researchers to train significantly deeper residual networks (Sankararaman et al., 2019), and these networks have achieved state-of-the-art performance on a number of tasks.

However batch normalization also has a number of limitations. It breaks the independence between training samples in a minibatch, which makes it harder to apply in certain models (Girshick, 2015), and also contradicts the assumptions of most theoretical models of optimization (Polyak & Juditsky, 1992; Mandt et al., 2017; Park et al., 2019). The normalization operation itself can constitute a significant fraction of the total cost of computing a parameter update in some models (Wu et al., 2018). Batch normalization also performs poorly when the batch size is too small (Ioffe, 2017; Wu & He, 2018), which often limits the size of model that can be trained on a single device.

**Fixup:** Zhang et al. (2019) proposed Fixup initialization which removes batch normalization entirely from residual networks. They demonstrated that this algorithm is able to train ResNet-50 to competitive validation accuracies on ImageNet, while it also displayed promising results when training very deep (1000+ layers) Wide-ResNets on CIFAR-10. The Fixup algorithm has four components:

1. The classification layer and final convolution of each residual branch are initialized to zero.
2. The initial weights of the remaining convolutions are scaled down by $L^{-1/(2m-2)}$, where $L$ is the number of residual branches and $m$ the number of convolutions in each branch.
3. A scalar multiplier is introduced at the end of each residual branch, intialized to one.
4. Scalar biases are introduced before every convolution, linear and element-wise activation layer, initialized to zero.

It is not clear whether all of these components are required to enable efficient training without batch normalization, or if a subset of these components would be sufficient. The authors claim that component 2 above is crucial, however we will demonstrate below that it is not required in practice. Additionally, the authors only provide results at a single batch size and do not tune the learning rate. We demonstrate below that a significantly simpler initialization scheme (ZeroInit) enables the efficient training of very deep residual networks without batch normalization when the batch size is not too large. ZeroInit and Fixup both achieve similar test accuracies to batch normalization when the batch size is small, but they underperform batch normalization when the batch size is very large.

## 3 THE BENEFITS OF BATCH NORMALIZATION

The primary goal of this paper is to identify the empirical benefits of batch normalization when training residual networks. We perform most of our experiments on CIFAR-10 (Krizhevsky et al., 2009), since this enables us to run thorough experiments, averaging across multiple training runs and tuning the learning rate at each batch size. To verify that our conclusions scale to larger datasets, we provide a limited set of experiments on ImageNet in section 5 (Russakovsky et al., 2015).

**Experimental Setup:** In figure 1, we provide the mean performance of a 16-4 Wide-ResNet (Zagoruyko & Komodakis, 2016), trained on CIFAR-10 for 200 epochs at a range of batch sizes. At each batch size, we train the network 15 times for a range of learning rates on a logarithmic grid, and we independently measure the mean and standard deviation of the best 12 runs on both the training loss and the test accuracy. This ensures that our results are not corrupted by outliers or failed runs. Throughout this paper, we train using SGD with heavy ball momentum, and fix the momentum coefficient $m = 0.9$. The optimal test accuracy is the mean performance at the learning rate whose mean test accuracy was highest, while the optimal training loss is the mean training loss at the learning rate whose mean training loss was lowest. We always verify that the optimal learning rates are not at the boundary of our grid search. Although we tune the learning rate on the test set, we emphasize that our goal is not to achieve state of the art results. Our goal is to compare the performance of different training procedures. We apply the same experimental protocol in each case.

We adopt the following simple learning rate schedule. We first hold the learning rate constant for 100 epochs, before dropping the learning rate by a factor of 2 every 10 epochs. We find that this simple schedule outperforms the original sequence of 3 drops proposed by He et al. (2016a). We apply data augmentation including padding, random crops and left-right flips and we also use L2 regularization with a coefficient of $5 \times 10^{-4}$. We provide results both with and without batch normalization. When using batch normalization, we evaluate the batch statistics over a fixed ghost batch size of 64 (Hoffer et al., 2017), before accumulating gradients to form larger batches. Evaluating the batch statistics over a fixed number of training examples independent of the batch size preserves the benefits of noise when the batch size is large (Luo et al., 2019), and reduces communication overheads in distributed

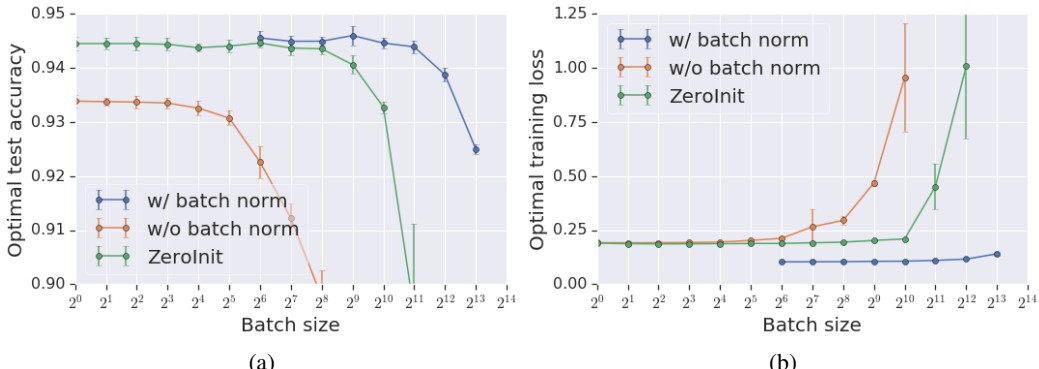

Figure 1: A 16-4 Wide-ResNet, trained on CIFAR-10 for 200 epochs at a range of batch sizes. We report the performance both with and without ghost batch normalization, and we also provide the performance without ghost batch normalization using our modified initialization scheme "ZeroInit". We perform a grid search to identify the optimal learning rate which maximizes the test accuracy at each batch size in figure 1a, and which minimizes the training loss at each batch size in figure 1b, and we report the mean performance of the best 12 of 15 runs. a) Batch normalization has at least two benefits. It improves the final test accuracy, and it also substantially increases the largest batch size we can use efficiently during training. Our modified initialization scheme recovers most of the improvement in the test accuracy, however the largest efficient batch size remains an order of magnitude smaller. b) Batch normalization achieves smaller training losses at all batch sizes, while both methods without normalization achieve similar training losses when the batch size is small.

training (we provide experiments using the original formalism of batch normalization in figure 3). For completeness, we also provide results here without normalization using a simple modification to the initialization scheme, "ZeroInit". We will describe this modification in full in section 4.

**Results:** Looking at the optimal test accuracy in figure 1a, we make three observations. First, the optimal test accuracy is substantially higher with batch normalization than without batch normalization, even after we tune both the learning rate and the batch size. Second, both with and without batch normalization, the performance is independent of batch size in the small batch limit, before beginning to fall as the batch size rises. Third, the largest batch size at which one can train efficiently with batch normalization is substantially larger than the corresponding batch size without batch normalization. Since the number of training epochs is fixed, this indicates that one can train Wide-ResNets with significantly fewer parameter updates when using batch normalization. Given access to sufficient hardware, this will enable practitioners to dramatically reduce the wallclock time of training (Goyal et al., 2017). ZeroInit significantly improves the optimal test accuracy achieved without normalization, and is competitive with batch normalization (within one standard deviation) for batch sizes $B \lesssim 256$. However it underperforms batch normalization when the batch size is large. Now considering the optimal training accuracy in figure 1b, we find that batch normalization achieves lower optimal training losses at all batch sizes, and continues to perform well when the batch size is large. The optimal training loss without batch normalization is similar both with and without ZeroInit. However when ZeroInit is not used, the optimal training loss degrades rapidly as the batch size is increased.

To help interpret these results, in figure 2 we provide the optimal learning rates at which both the training loss is minimized and the test accuracy is maximized. From figure 2a, we conclude that for all three techniques, the optimal learning rate is initially proportional to the batch size, before saturating when the batch size is large (Krizhevsky, 2014; Goyal et al., 2017; Smith et al., 2017; McCandlish et al., 2018; Shallue et al., 2018). Furthermore, the optimal ratio of the learning rate to the batch size is similar for all three techniques. However, while the optimal learning rate saturates at a maximum value of $2^{-3}$ without batch normalization, it reaches a larger value of $2^1$ with ZeroInit, and an even larger value of $2^4$ with batch normalization. This directly explains why batch normalization and ZeroInit perform better when the batch size is large; they have larger maximum stable learning rates which enables them to scale efficiently to larger batches (McCandlish et al., 2018). We observe very similar results when considering the optimal learning rates minimizing the training loss in figure 2b.

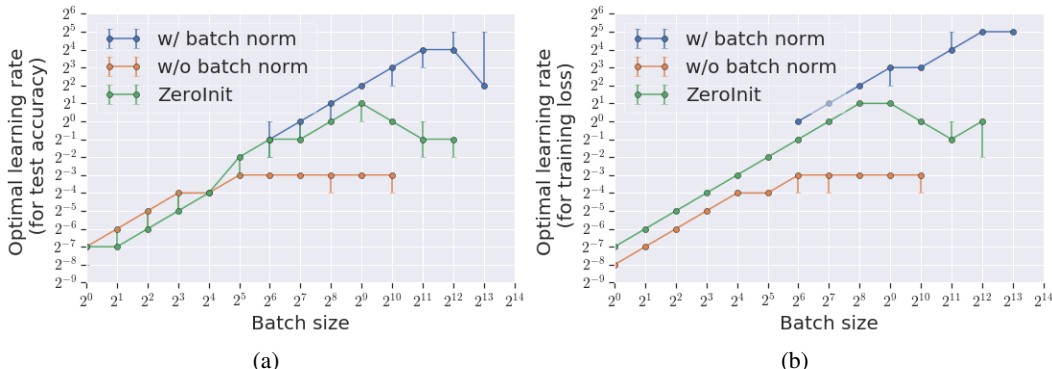

Figure 2: A 16-4 Wide-ResNet, trained on CIFAR-10 for 200 epochs at a range of batch sizes. a) For all three methods, the learning rate which maximizes the test accuracy is proportional to the batch size when the batch size is small, before saturating when the batch size is large. Batch normalization achieves significantly larger optimal learning rates in the large batch limit, however all three methods have similar optimal learning rates at the same batch size when the batch size is small. This demonstrates that the stability of batch normalization at large learning rates is not the reason why batch normalization performs better on the test set when the batch size is small. b) We observe similar trends when we evaluate the optimal learning rates which minimize the training loss.

**Discussion:** Santurkar et al. (2018) and Bjorck et al. (2018) argued that the empirical success of batch normalization is primarily explained by the increase in the maximum stable learning rate, which arises because batch normalization improves the conditioning of the loss landscape. However our results conclusively demonstrate that this is not the case. While a larger maximum stable learning rate is the primary reason why batch normalization performs well with large training batches, when the batch size is small the optimal learning rate both with and without batch normalization is also small. Nevertheless, the optimal test accuracy and the optimal training loss are significantly improved when batch normalization is used in the small batch limit as well as the large batch limit.

We conclude that batch normalization increases the largest stable learning rate, and this enables us to reduce the number of parameter updates required to train the model when training with large batches. However batch normalization also improves the final training loss and test accuracy, even when the batch size and learning rate are small. Our results demonstrate that one cannot understand batch normalization without studying its performance at a range of batch sizes. ZeroInit recovers much of the performance gap on the test set when the batch size is small, but it does not close the gap in the training loss, and it performs worse with large batches. It is difficult to measure exact wallclock times in our distributed training setup, but we find that 16-4 wide residual networks typically complete 200 epochs *roughly* 40% *faster* at batch size 64 without batch normalization (Wu et al., 2018).

## 4 "ZeroInit": a simple alternative to batch normalization

We established above that larger stable learning rates are not the only benefit of batch normalization, and that they do not explain why batch normalization achieves lower training losses and test accuracies under a finite epoch budget when the batch size is small (Santurkar et al., 2018; Bjorck et al., 2018). We now identify the minimal components required to train very deep ResNets without normalization. Balduzzi et al. (2017) and Yang et al. (2019) both argued that batch normalization when used with skip connections preserves the correlations between gradients at initialization, and Balduzzi et al. (2017) argued that these correlations can also be preserved by initializing deep networks close to linear functions. Inspired by these results, we introduce a new initialization scheme, "ZeroInit", comprising three simple components

1. We introduce a scalar multiplier at the end of each residual branch, initialized to zero.

2. We introduce biases to each convolution and the classification layer, initialized to zero.

3. We apply Dropout on the classification layer. If not specified, the drop probability is 60%.

Table 1: The performance of a range of 16-4 Wide-ResNets on CIFAR-10 without batch normalization at batch size 64. We use a compute budget of 200 epochs and follow our standard learning rate schedule. We perform two independent grid searches to identify the learning rate which minimizes the training loss and the learning rate which maximizes the test accuracy, and we provide the mean performance of the best 12 out of 15 runs. Introducing a scalar multiplier initialized at zero to each residual branch is sufficient to achieve a test accuracy of $94\%$, and this configuration also achieves the lowest training loss of $0.181$. Meanwhile ZeroInit, which combines this scalar multiplier with biases and dropout, improves the optimal test accuracy but also increases the training loss.

| Method | Optimal test accuracy (%) | Optimal training loss |
|---|---|---|
| 16-4 wide ResNet w/o batch normalization | $92.3 \pm 0.3$ | $0.212 \pm 0.004$ |
| + biases | $91.9 \pm 0.2$ | $0.205 \pm 0.003$ |
| + dropout | $93.2 \pm 0.1$ | $0.226 \pm 0.001$ |
| + scalar multiplier | $94.0 \pm 0.1$ | $\mathbf{0.181 \pm 0.001}$ |
| + biases & dropout | $93.6 \pm 0.1$ | $0.212 \pm 0.001$ |
| + ZeroInit (scalar multiplier, biases & dropout) | $\mathbf{94.4 \pm 0.1}$ | $0.189 \pm 0.001$ |

Table 2: The performance of a $d$-2 Wide-ResNet on CIFAR-10 at batch size 64, for a range of depths $d$. We use a compute budget of 200 epochs and follow our standard learning rate schedule. We perform a grid search to identify the optimal learning rate, and provide the mean performance of the best 12 out of 15 runs. Both batch normalization and ZeroInit are able to train very deep networks up to (at least) one thousand layers. The optimal learning rate is only weakly dependent on depth.

| | Batch Normalization | | ZeroInit | |
|---|---|---|---|---|
| Depth | Test accuracy (%) | Optimal learning rate | Test accuracy (%) | Optimal learning rate |
| 16 | $93.5 \pm 0.1$ | $2^{-1}$ | $93.3 \pm 0.1$ | $2^{-2}$ |
| 100 | $94.7 \pm 0.1$ | $2^{-1} \ (2^{-2} - 2^0)$ | $94.2 \pm 0.1$ | $2^{-2}$ |
| 1000 | $94.6 \pm 0.1$ | $2^{-2} \ (2^{-3} - 2^0)$ | $94.3 \pm 0.2$ | $2^{-2} \ (2^{-3} - 2^{-1})$ |

We note that each of these modifications should not require changing more than a couple of lines of code. Modification 1 makes the residual blocks linear at initialization, so long as there are no non-linear activations on the skip connection. This preserves the gradient correlations at initialization (Balduzzi et al., 2017), and we show below that this single change is sufficient to train very deep ResNets without normalization. Modification 2 is added to preserve the model expressivity, and Modification 3 is added to recover some of the additional regularization benefits batch normalization provides (Luo et al., 2019). We note that neither Wide-ResNets (Zagoruyko & Komodakis, 2016) nor ResNet-V2 (He et al., 2016b) have non-linear activations on the skip connection, while ResNet-V1 (He et al., 2016a) does. We discuss how to modify ZeroInit for ResNet-V1 in section 5. We note that Fixup initialization also initializes the residual branches at zero, while Goyal et al. (2017) proposed initializing the scalar multiplier in the final batch normalization layer of the residual branch to zero.

We provide an ablation study in table 1, which demonstrates that solely introducing a scalar multiplier initialized to zero is sufficient to train our 16-4 wide residual network to $94\%$ test accuracy. This configuration also achieves the lowest training loss. Introducing biases and dropout further increases the test accuracy to $94.4\%$. In table 2, we compare the performance of batch normalization and ZeroInit at a range of depths between 16 and 1000 layers. To ensure each model fits on a single GPU, we reduce the width factor from 4 to 2, which slightly reduces the test accuracy. While

Table 3: The performance of a $d$-2 Wide-ResNet on CIFAR-10 at batch size 64, for a range of depths $d$. We train for 200 epochs. We perform a grid search to identify the optimal learning rate, and provide the mean performance of the best 12 out of 15 runs. L2 regularization is not required to train very deep networks with ZeroInit, and we can also train very deep networks without biases or dropout, using solely a scalar multiplier at the end of each residual branch initialized to zero.

| | ZeroInit without L2 | | Scalar Multiplier | |
|---|---|---|---|---|
| Depth | Test accuracy (%) | Optimal learning rate | Test accuracy (%) | Optimal learning rate |
| 16 | $91.0$ | $2^{-3}$ | $93.1 \pm 0.1$ | $2^{-2} \ (2^{-3} - 2^{-2})$ |
| 100 | $92.2 \pm 0.1$ | $2^{-3}$ | $94.1 \pm 0.1$ | $2^{-2} \ (2^{-3} - 2^{-2})$ |
| 1000 | $92.5 \pm 0.2$ | $2^{-3}$ | $94.2 \pm 0.1$ | $2^{-1} \ (2^{-2} - 2^{-1})$ |

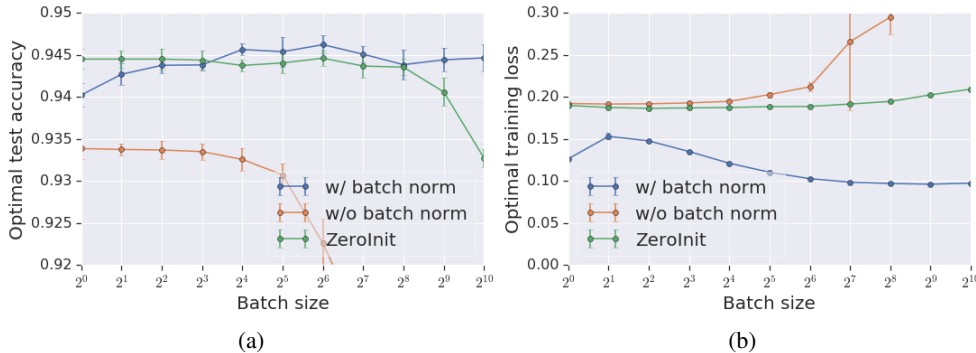

(a)                                                       (b)

Figure 3: The performance of a 16-4 Wide-ResNet in the small batch limit, trained on CIFAR-10 for 200 epochs. We perform a grid search over the learning rate and provide the mean performance of the best 12 out of 15 runs. In contrast to the rest of this paper, we do not use ghost batch normalization, and evaluate the batch statistics over the whole batch. Batch normalization outperforms ZeroInit on the training loss. However it underperforms ZeroInit on the test set when the batch size is very small.

batch normalization slightly outperforms ZeroInit for all depths, both techniques are able to train Wide-ResNets with one thousand layers to over $94\%$ test accuracy within 200 epochs. Intriguingly, we also find that in both cases the optimal learning rate only depends weakly on the network depth.

When increasing the network depth, we found that the loss at initialization becomes increasingly dominated by the L2 regularization loss, and we suspected that this may explain why ZeroInit is able to train very deep networks without rescaling the convolutional layers (as proposed by Zhang et al. (2019)). Therefore in table 3, we also provide the test accuracy when training very deep wide residual networks using ZeroInit without L2 regularization. To our surprise, we found that training is still stable, and that the optimal learning rate remains independent of depth. We therefore conclude that rescaling is not necessary in practice. We also verify that it is possible to train very deep networks solely by introducing the scalar multiplier in ZeroInit to each residual branch initialized to zero.

Finally, we note that a number of authors are interested in alternatives to batch normalization for very small batch training (Wu & He, 2018). In figure 3, we compare the performance of ZeroInit and batch normalization, when training a 16-4 Wide-ResNet for 200 epochs at a range of batch sizes $1 \leq B \leq 1024$. In contrast to the rest of this paper, we do not apply ghost batch normalization here (Hoffer et al., 2017), and instead evaluate the batch statistics over the training minibatch. As expected, the final performance of ZeroInit is independent of the batch size in the small batch limit $B \lesssim 256$ (within one standard deviation), but the test accuracy achieved by batch normalization drops for batch sizes $B \lesssim 64$. ZeroInit may therefore provide an alternative to batch normalization in small batch settings. ZeroInit still significantly underperforms batch normalization for large batch sizes $B \gtrsim 256$. We provide the corresponding optimal learning rates at each batch size in appendix A.

## 5 COMPARISONS ON IMAGENET

We now compare batch normalization, Fixup initialization and ZeroInit on ImageNet using ResNet-50. We compare performance on both the original ResNet-50-V1 architecture (He et al., 2016a), and the modified ResNet-50-V2 (He et al., 2016b). Like Wide-ResNets, ResNet-50-V2 contains an identity skip connection through all the residual blocks. Meanwhile ResNet-50-V1 introduces a ReLU at the end of every residual block after the residual branch and skip connection merge. While V2 is thought to perform better in most cases and is easier to train (He et al., 2016b), V1 is still often used as a benchmark. Since we wish to compare networks with and without an identity skip connection, we train using the original versions of ResNet-50-V1 and ResNet-50-V2 (He et al., 2016a;b). We do not use the modifications to ResNet-50-V1 described by Goyal et al. (2017), although these are known to improve the validation accuracy by 1-2%. We use a ghost batch size of 256 (Hoffer et al., 2017).

In table 4, we present the performance of batch normalization, Fixup and ZeroInit when training ResNet-50-V2 (He et al., 2016b) on ImageNet at a range of batch sizes. We train for 90 epochs,

Table 4: The performance of ResNet-50-V2 on ImageNet. We use a compute budget of 90 epochs and perform a grid search to identify the optimal learning rate which maximizes the top-1 validation accuracy. We perform a single run at each learning rate and report both top-1 and top-5 accuracy scores. We use a drop probability of 0.2 for ZeroInit. Both Fixup and ZeroInit are competitive with batch normalization at small batch sizes, while batch normalization is better at larger batch sizes.

| Batch Size | Batch Normalization | Fixup | ZeroInit without Dropout | ZeroInit |
|---|---|---|---|---|
| 256 | 74.97 / 92.20 | 74.81 / 91.84 | 74.93 / 91.89 | 75.56 / 92.35 |
| 1024 | 74.93 / 92.13 | 74.60 / 91.69 | 74.61 / 91.81 | 75.46 / 92.53 |
| 4096 | 74.88 / 91.93 | 73.02 / 90.59 | 70.76 / 89.15 | 72.65 / 90.70 |

Table 5: The performance of ResNet-50-V1 on ImageNet. We use a compute budget of 90 epochs and perform a grid search to identify the optimal learning rate which maximizes the top-1 validation accuracy. We perform a single run at each learning rate and report both top-1 and top-5 accuracy scores. We use a drop probability of 0.2 for ZeroInit. Fixup performs well when the batch size is small, but significantly underperforms batch normalization when the batch size is large. ZeroInit performs poorly at all batch sizes, but its performance improves considerably if we add a scalar bias before the final ReLU in each residual block (after the skip connection and residual branch merge).

| Batch Size | Batch Normalization | Fixup | ZeroInit | ZeroInit + Scalar Bias |
|---|---|---|---|---|
| 256 | 75.57 / 92.48 | 74.41 / 91.65 | 70.05 / 89.16 | 75.23 / 92.37 |
| 1024 | 75.32 / 92.37 | 74.43 / 91.74 | 68.40 / 87.82 | 74.94 / 91.99 |
| 4096 | 75.37 / 92.38 | 72.37 / 90.28 | 68.19 / 87.92 | 70.82 / 89.60 |

and hold the learning rate constant for 45 epochs before decaying the learning rate by a factor of 2 every 5 epochs. We find that both Fixup and ZeroInit are competitive with batch normalization when the batch size is small, but both underperform batch normalization when the batch size is large. These results match our earlier observations when training wide residual networks in section 3. The performance of Fixup is comparable to the performance of ZeroInit without dropout.

In table 5, we present the performance of batch normalization, Fixup and ZeroInit when training Resnet-50-V1 (He et al., 2016a). As discussed above, and unlike both ResNet-V2 and Wide-ResNets (Zagoruyko & Komodakis, 2016), this network introduces a ReLU at the end of the residual block after the skip connection and residual branch merge. We find that Fixup slightly underperforms batch normalization when the batch size is small, but considerably underperforms batch normalization when the batch size is large. Meanwhile ZeroInit significantly underperforms both batch normalization and Fixup at all batch sizes. This is not surprising, since we designed ZeroInit for models which contain an identity skip connection through the residual blocks. We also provide additional results for a modified version of ZeroInit, which additionally contains a single scalar bias in each residual block, just before the final ReLU (after the skip connection and residual branch merge). We find that this scalar bias eliminates the gap in validation accuracy between Fixup and ZeroInit when the batch size is small. We conclude that only two components of Fixup are essential; initializing the residual branch at zero, and introducing a scalar bias after the skip connection and residual branch merge.

## 6 CONCLUSIONS

Batch normalization brings at least two practical benefits in residual networks. It increases the test accuracy, and it also substantially increases the largest stable learning rate which can be used during training. This second benefit enables efficient training with larger batch sizes, which can be exploited to reduce the wallclock time of training. However the increase in the largest stable learning rate cannot be the cause of the increase in the test accuracy, since we achieve similar test accuracies with smaller learning rates when the batch size is smaller. This demonstrates that empirical studies of batch normalization must consider a range of batch sizes. We also identify a simple modification to the initialization scheme, "ZeroInit", which allows us to train very deep (1000+ layer) residual networks without normalization. This scheme achieves similar test accuracies to batch normalization for reasonable batch sizes and outperforms batch normalization for very small batch sizes, but it underperforms batch normalization when the batch size is large. ZeroInit achieves similar validation accuracy to batch normalization when training ResNet-50-V2 on ImageNet at batch size 1024.

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

## A    THE OPTIMAL LEARNING RATES IN THE SMALL BATCH LIMIT

In figure 4, we provide the optimal learning rates when training a 16-4 Wide-ResNet for 200 epochs on CIFAR-10 in the small batch size limit. We evaluate the batch statistics over the full training minibatch, which enables us to consider the limit $B \to 1$ when using batch normalization. We note that the test accuracy and training loss at each batch size was provided in the main text in figure 3. Taken together, these two plots provide further evidence that larger stable learning rates do not explain why batch normalization performs well on the training set and the test set when the batch size is small.

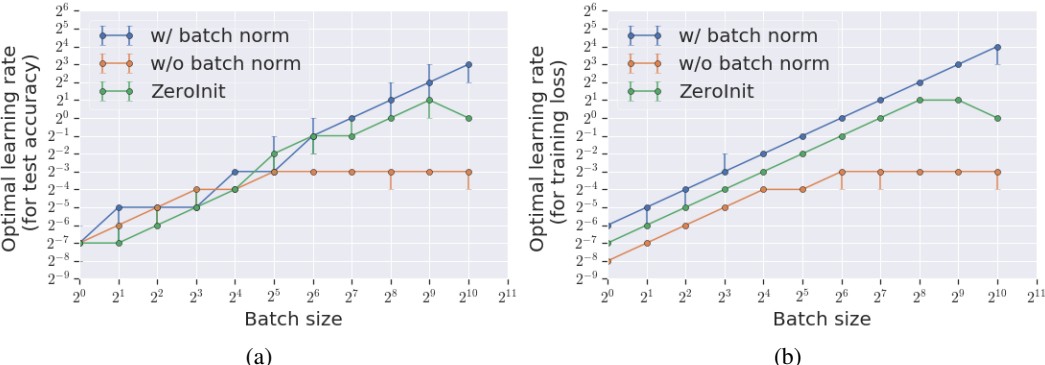

(a)                                                      (b)

Figure 4: A 16-4 Wide-ResNet, trained on CIFAR-10 for 200 epochs at a range of batch sizes in the small batch limit. In order to study batch normalization as the batch size $B \to 1$, we use the original formulism of batch normalization here, where the batch statistics are evaluated over the training minibatch. a) The learning rate which maximizes the test accuracy is proportional to the batch size when the batch size is small, and all three methods exhibit similar learning rates at the same batch size in this limit. This demonstrates that the stability of batch normalization at large learning rates is not the reason why batch normalization performs better on the test set when the batch size is small. b) We observe similar trends when we evaluate the optimal learning rates which minimize the training loss.

