# OpenReview forum: "Batch Normalization has Multiple Benefits: An Empirical Study on Residual Networks"
_ICLR.cc/2020/Conference — Reject_

### Official Review · AnonReviewer1 · 2019-10-23
**Official Blind Review #1**

**Rating:** 1

**Review:**

This paper conducts extensive experiments to verify two practical benefits of batch normalization. i) It increases the final test accuracy and the largest stable learning rate; ii) it enables efficient training with larger batches and a larger learning rate. In addition, the authors propose a new initialization scheme, “ZeroInit”, to train a deep ResNet to improve the test accuracy. My detailed comments are as follows.


Positive points:

1. The experiments are sufficient. In this paper, the authors conduct extensive experiments to explore the benefits of batch normalization, and verifies the effectiveness of the proposed “ZeroInit”.

2. The method is effective in some cases. Specifically, the proposed “ZeroInit” outperforms batch normalization when the batch size is small, and it is competitive with batch normalization when the batch size is not too large.

Negative points:

1. The importance and novelty of the empirical study should be emphasized. The practical benefits of batch normalization can be also found in other papers. For the first benefit, most studies (Bjorck et al. 2018) have found that batch normalization is able to improve the test accuracy. For the second benefit, batch normalization requires a large batch size and a large learning rate (Santurkar et al., 2018). Therefore, what is the difference between this paper and others? More critically, it is necessary to explain why batch normalization has these benefits. It would be better to provide empirical or theoretical justifications to support these.

2. The motivation of the proposed “ZeroInit” is not clear. (Balduzzi et al., 2017) states that “the correlations can be preserved by initializing deep networks close to linear functions”. It is not clear how “ZeroInit” preserves the correlations?

3. Why initialize the scalar multiplier and biases to zero? What are the benefits of the zero initialization? Actually, the scalar multiplier and biases can be randomly initialized. When they are randomly initialized, what is the performance of the initialization? It is an important baseline to justify the effectiveness of the proposed initialization method.

4. The technical details of “ZeroInit” are not clear. It would be better to express the proposed initialization “ZeroInit” in the mathematical formulation.

5. The proposed initialization “ZeroInit” is designed for deep ResNets. How to extend it to the other deep neural networks?

6. This paper states that “the empirical success of batch normalization …improves the conditioning of the loss landscape. However, our results conclusively demonstrate that this is not the case”. Does it mean that batch normalization does not improve the conditioning of the loss landscape? However, the empirical results cannot justify this statement and explain the success of batch normalization.

7. Some results of the figures are missing. In Figure 1, the experimental results of w/o batch norm with varying batch sizes (2^0 ~ 2^5) are missing. Similarly, Figure 2 also has missing results. Please provide more discussions about these missing results.


**Experience Assessment:**

I have published one or two papers in this area.

**Review Assessment: Checking Correctness Of Derivations And Theory:**

I carefully checked the derivations and theory.

**Review Assessment: Checking Correctness Of Experiments:**

I carefully checked the experiments.

**Review Assessment: Thoroughness In Paper Reading:**

I read the paper thoroughly.

---

> ### Author Response · Authors · 2019-11-06
> **Response to review**
>
> We thank the reviewer for their comments. However their review also contains a number of misunderstandings regarding our work (which we will address when we update the manuscript). We hope the reviewer might reconsider their score if we clarify our contributions here.
>
> Bjorck et al. and Santurkar et al. both claim that the primary benefit of batch normalization which explains its superior performance is that it improves the conditioning of the loss, enabling stable training with larger learning rates. We show empirically that this statement is false. Batch normalization does enable larger learning rates, and this explains why it is possible to efficiently train batch normalized networks with larger batch sizes. However when the batch size is small, the optimal learning rate both with and without batch normalization is also small, yet batch normalization continues to significantly increase the test accuracy and reduce the training loss.
>
> In addition, we propose a simple and theoretically motivated initialization scheme, ZeroInit, which enables us to train very deep residual networks to high test accuracy without any normalization. This scheme is similar to Fixup initialization but it is significantly simpler to implement and is based on clear theoretical principles. We also demonstrate that many components of Fixup which the authors claim are essential are in fact unnecessary (most notably the rescaling of conv layers at initialization).
>
> The authors of Fixup initialization also claimed that Fixup initialized networks could be trained at the same large learning rates as batch normalized networks. We show that this claim is false. Unlike batch normalization, both ZeroInit and Fixup cannot be trained with very large learning rates. Consequently, both schemes are competitive with batch normalization for small/moderate batch sizes but both underperform batch normalization when the batch size is large.
>
> We believe these contributions will be very valuable to the ML community. In response to the specific negative points raised:
>
> 1. See the discussion of our contributions above. Our paper demonstrates that a key claim of both Bjorck et al. and Santurkar et al. is false; stable training with large learning rates explains why batch normalized networks can be trained with large batch sizes but it does not explain why batch normalization significantly increases the test accuracy and reduces the training loss when the batch size is small. Although our results are primarily empirical, the success of ZeroInit strongly suggests that one of the key benefits of batch normalization in residual networks is to preserve gradient correlation, as proposed by Balduzzi et al..
>
> 2. Introducing a scalar multiplier to the residual branch initialized at zero ensures that, at initialization, the signal only propagates through the skip connection and therefore the residual block computes an identity function (trivially a linear function). This ensures that the network at initialization is close to linear, preserving the gradient correlations.
>
> 3. If we did not initialize the scalar multipliers at zero, the residual block would not compute the identity function at initialization, and therefore the gradient correlations would not be preserved. It is not necessary to initialize the biases at zero, although this is common practice. We will be happy to add an additional ablation study exploring this topic to the text.
>
> 4. We will be happy to try to clarify this in the updated version. However, we feel that the definition of ZeroInit at the bottom of page 5 is sufficiently clear for future authors to implement.
>
> 5. Like Fixup, ZeroInit is designed for ResNets, and it cannot be trivially extended to other networks. However it does suggest a simple guiding principle for ensuring that deep networks are trainable at initialization, namely that one should ensure that networks are randomly initialized at the boundary between linear and nonlinear functions. For instance, Xiao et al. [1] found that one can train very deep convolutional networks without batch normalization by choosing an initialization scheme with this property. We can clarify this in an updated version of the paper.
>
> 6. As clarified above, we did not claim that batch normalization does not improve the conditioning of the loss. Batch normalization does improve the conditioning of the loss, however our results show that this does not explain why batch normalization significantly increases the test accuracy and reduces the training loss when the batch size is small.
>
> 7. As stated in the text, in Figures 1 and 2 we use ghost batch normalization (Hoffer et al.), whereby the batch statistics are estimated over 64 examples. Consequently we cannot reduce the batch size below 64. However in Figures 3 and 4 we estimate the batch statistics over the full batch size, and we are able to reduce the batch size to 1.
>
> [1] Dynamical Isometry and a Mean Field Theory of CNNs, ICML 2018

---

### Official Review · AnonReviewer3 · 2019-10-23
**Official Blind Review #3**

**Rating:** 3

**Review:**

The name "ZeroInit" is very confusing, because that is how FixUp was called initially https://openreview.net/forum?id=H1gsz30cKX , perhaps the authors should consider a different name. I will call it "NewZeroInit" in my review to avoid confusion.

The paper focuses on training image classification networks without batch normalization. The authors claim that effectiveness of batch normalization, and methods which attempt to eliminate it, should be tested on a wide range of learning rates. On experiments performed on CIFAR they find that batch normalization is able to achieve high accuracy even with very high learning rates, in line with Goyal et al. 2017. Based on this, they propose a simplification of FixUp for image classification, in which they remove the need in progressive scaling of initialization, and propose to remove weight decay regularization, while adding dropout on the last layer. This "NewZeroInit" is tested on ImageNet and compares favorably to batch normalization and FixUp.

The closest studies are FixUp and Goyal et al. 2017, with the difference that FixUp studies both image classification ResNet and seq2seq approaches in the absence of batch normalization, and Goyal et al. show a wide range of large scale experiments on full scale ImageNet, whereas "NewZeroInit" studies small scale CIFAR dataset. It is thus unclear if "NewZeroInit" transfers to seq2seq.
There is also "Bag of Tricks for Image Classification with Convolutional Neural Networks" by He et al 2018 (missing citation) which evaluates a similar set of tricks on ImageNet ResNet-50 with batch normalization. In particular, they show that removing weight decay from BN bias and setting scaling gamma to 0 initially significantly improves the results.

On page 4 the authors say "Given access to sufficient hardware, this will enable practitioners to dramatically reduce wallclock time of training (Goyal et al.)". It is not clear what they mean, since Goyal et al. already enabled the reduction by increasing learning rate and minibatch size on ImageNet, whereas the results authors show are on small CIFAR dataset.

On page 5 the authors mention that they introduce bias to each convolution and classification layer, which is surprising because it is a standard way to composing a convolutional network.

Overall, the most significant contributions of the paper are:
 - a study of minibatch size on CIFAR
 - removing weight decay from FixUp on ImageNet

Also, I am interested in the following results:
 - clear comparison of FixUp with "NewZeroInit" for image classification
 - ImageNet ResNet-50 results with dropout regularization in the final layer
 - ImageNet ResNet-50 results with FixUp, dropout regularization and no weight decay.
 - (optionally) seq2seq with NewZeroInit instead of FixUp.

Without these results it hard to judge the novelty and contributions of the paper, so I propose reject.

**Experience Assessment:**

I have published in this field for several years.

**Review Assessment: Checking Correctness Of Derivations And Theory:**

I assessed the sensibility of the derivations and theory.

**Review Assessment: Checking Correctness Of Experiments:**

I assessed the sensibility of the experiments.

**Review Assessment: Thoroughness In Paper Reading:**

I read the paper at least twice and used my best judgement in assessing the paper.

---

> ### Author Response · Authors · 2019-11-06
> **Response to review**
>
> We thank the reviewer for their comments on our work.
>
> First, we would like to clarify that we do not remove weight decay from Fixup. We train with weight decay throughout the paper, and we provide the decay coefficients used for each experiment. The only experiment for which we removed weight decay is the ablation study in table 3. This ablation study confirms that the rescaling of conv layers proposed in Fixup is not required to train very deep residual networks without batch normalization, directly contradicting claims made in the Fixup paper. We provided this ablation because the loss function of very deep networks at initialization is dominated by the L2 loss, and we were concerned that this may implicitly rescale the parameters in the conv layers early in training, even though ZeroInit does not rescale these parameters explicitly.
>
> Second, we emphasize that a major contribution of this work is to study the benefits of batch normalization empirically. Bjorck et al. and Santurkar et al. (both NeurIPS 2018) claimed that the key benefit of batch normalization is to improve the conditioning of the loss and increase the largest stable learning rate. Our results show this claim is false. When the batch size is small, the optimal learning rate both with and without batch normalization is small, yet batch normalization still significantly increases the test accuracy and reduces the training loss. Batch normalization does enable larger learning rates, but this is only beneficial when the batch size is large.
>
> Finally, we introduced ZeroInit, which is significantly simpler than Fixup, well motivated by theory and achieves the same performance on ImageNet. Furthermore, the authors of Fixup initialization claim that Fixup enables stable training at the same large learning rates achieved by batch normalization, but we show that this is not true. Both ZeroInit and Fixup are not stable with very large learning rates, and consequently both are only competitive with batch normalization for small/moderate batch sizes (eg < 1000 on ImageNet).
>
> We now address some of the additional points brought up in the review below.
>
> 1. The reviewer mentions multiple times that we only provide a study of minibatch size on CIFAR-10. However we also provide experiments at a range of batch sizes between 256 and 4096 on ImageNet in tables 4 and 5. These results verify that ImageNet follows the same trends we observed on CIFAR-10.
>
> 2. “Given access to sufficient hardware, this will enable practitioners to dramatically reduce wallclock time of training (Goyal et al.)”: Our point is that methods like batch normalization which enable larger learning rates are particularly useful when one wishes to minimize the wallclock time of training, since one can increase both batch size and learning rate, and then parallelize computation over multiple GPUs. If large learning rates were not stable, Goyal et al. would not have been able to increase the learning rate and batch size to reduce the wall clock time.
>
> 3. Biases are often used in conv layers, but these are usually removed when batch normalization is used. When replacing batch normalization with ZeroInit, we simply add these biases back into the network. As we show in the ablation study in table 1, these added biases only bring marginal benefits while the scalar multiplier initialized at zero is essential. We note that the simplicity of ZeroInit is a key positive contribution of our work.
>
> 4. The reviewer asks for a proper evaluation of Fixup and ZeroInit for image classification. However, we already provide a thorough comparison of Fixup and ZeroInit on ImageNet at a range of batch sizes in tables 4 and 5 for both ResNet50-V1 and ResNet50-V2. We find that ZeroInit outperforms Fixup when the batch size is small but slightly underperforms Fixup when the batch size is large.
>
> 5. The reviewer also asks for ImageNet results for Fixup with dropout. We note that in table 4, we provided ImageNet results for ZeroInit without dropout in order to enable a fair comparison to Fixup without additional regularization. As we stated in the text, ZeroInit without dropout performs similarly to both Fixup and batch normalization when the batch size is small. That said, we will run additional experiments on ImageNet with Dropout for both batch normalized networks and Fixup initialization and add these to the text.
>
> 6. We were not aware before submission that Fixup had originally been called ZeroInit. We would be willing to change the name of the method to avoid confusion.
>
> 7. We will add a citation to “Bag of Tricks for Image Classification with Convolutional Neural Networks”. As clarified above, we did not remove weight decay from our networks, although we did confirm in an ablation study that weight decay is not required. We note that we did mention on page 6 that Goyal et al. set the scalar multiplier inside batch normalization to zero at initialization at the end of the residual branch.

---

> > ### Author Response · Authors · 2019-11-15
> > **Additional comparisons with dropout**
> >
> > We have performed the additional experiments requested by the reviewer. Please find below the comparisons between batch normalization, Fixup and ZeroInit, both with and without dropout. The experiments presented are for ImageNet classification with ResNet50-V2. When using dropout, we use a drop probability of 0.2 on the final classification layer for all methods.
> >
> >
> > Without dropout:
> >
> > Batch size     BatchNorm                        Fixup                  ZeroInit without Dropout
> >
> > 1024               74.93 / 92.13                 74.60 / 91.69                   74.61 / 91.81
> >
> >
> > With dropout:
> >
> > Batch size         BatchNorm w/ dropout            Fixup w/ dropout                 ZeroInit
> >
> > 1024                            74.82 / 91.98                          75.62 / 92.54                    75.46 / 92.53
> >
> >
> > These results seem to indicate that Fixup (like ZeroInit) does better with added regularization through dropout, and becomes comparable to ZeroInit at small batch sizes. Further, we see that batch normalization seems to do worse when dropout is added.
> >
> > Note that we independently tuned the learning rate for each of the experimental results shown above. We are currently in the process of evaluating these algorithms with dropout on other batch sizes.

---

### Official Review · AnonReviewer2 · 2019-11-07
**Official Blind Review #2**

**Rating:** 3

**Review:**

This paper conducts extensive experiments to study batch normalization, a very popular technique for training a deep convolutional network and its relationship with learning rate and batch size. In addition, the authors also propose a new initialization scheme, “ZeroInit”, to train a deep ResNet for better test accuracy. This is a very empirical study and the authors also show extensive experimental results. However, I do not see any novel findings in this study. Mostly this paper confirms the results of previous studies. The experimental results do not show much advantage of ZeroInit either. Overall, it is unclear what is the major novel contribution in this paper.

**Experience Assessment:**

I have read many papers in this area.

**Review Assessment: Checking Correctness Of Derivations And Theory:**

I assessed the sensibility of the derivations and theory.

**Review Assessment: Checking Correctness Of Experiments:**

I assessed the sensibility of the experiments.

**Review Assessment: Thoroughness In Paper Reading:**

I made a quick assessment of this paper.

---

> ### Author Response · Authors · 2019-11-07
> **Response to review**
>
> We thank the reviewer for their assessment of our work. The reviewer agrees that our results are extensive but is unclear what the major contributions of this paper are. To clarify:
>
> 1. The two most influential recent works studying the benefits of batch normalization are Bjorck et al. and Santurkar et al. (both NeurIPS 2018). Both papers argue that the key benefit of batch normalization is to improve the loss conditioning, which enables stable training with larger learning rates. Our experiments prove that this statement is false. When the batch size is small, the optimal learning rate with and without batch normalization is also small, yet batch normalized networks still achieve significantly higher test accuracies and lower training losses. Large learning rates cannot be the key benefit of batch normalization in residual networks.
>
> 2. There is great interest in finding alternatives to batch normalization. We propose an extremely simple initialization scheme, ZeroInit, which is competitive with batch normalization and can be trained without any normalization. The key component of ZeroInit is to add a scalar multiplier at the end of each residual branch initialized to zero. Note that this can be implemented in a single line of code.
>
> 3. ZeroInit is similar to the recently proposed Fixup initialization (Zhang et al., ICLR 2019). However, Zhang et al. argued that the key component of Fixup is to rescale the conv layers inside residual branches at initialization. We show empirically that this component is completely unnecessary, even if L2 regularization is also removed.
>
> 4. Zhang et al. also argued that Fixup is stable at the same large learning rates as batch normalization. Again, we show this claim is false. Both ZeroInit and Fixup are only stable at smaller learning rates and consequently they are both only competitive with batch normalization for small/moderate batch sizes (eg < 1000 on ImageNet)
>
> 5. Entire papers have been written whose sole purpose is to provide an alternative to batch normalization when the batch size is too small to estimate batch statistics. Examples include GroupNorm (over 250 citations) and batch renormalization (over 100 citations). ZeroInit can be trained at batch size 1 without any drop in final performance.
>
> In summary, we believe our work contains a number of valuable novel contributions. Crucially, our paper does not confirm the results of previous studies. Instead, it shows that the core claims in a number of highly influential papers are false empirically, while also proposing an alternative to batch normalization in residual networks which is significantly simpler to implement than existing methods.

---

### Public Comment · ~Antoine_Labatie1 · 2019-10-01
**A closely related work and a comment**

Hi,

I enjoyed reading your paper. That's an interesting work.

A closely related paper from ICML 2019 studied the influence of batch normalization and skip connections on the inductive bias of deep nets [1, 2]

I wanted to comment as well on the introduction of the scalar bias after the merging of layers for resnets v1. When batch normalization is used, my view is that at initialization each channel after the merging follows a random walk iteratively thresholded at 0. At high depth, this thresholded random walk becomes positive with high probability and the ReLU becomes the identity, thus easing training. This does not happen with ZeroInit without scalar biases. Is it possible to have your opinion on this view ? How do you initialize the scalar biases ?

[1] Characterizing Well-Behaved vs. Pathological Deep Neural Networks. ICML 2019.
[2] It’s Necessary to Combine Batch Norm and Skip Connections. Towards Data Science, 2019.

---

> ### Author Response · Authors · 2019-10-22
> **We initialize the additional scalar bias at zero**
>
> Hi Antoine,
>
> Our apologies for our slow reply, and thank you for your interest in our work! Your recent paper is very relevant and we will add a citation to it in future versions.
>
> We initialize the additional scalar biases in ResNet-V1 networks at zero. We have not studied in detail how these biases ease training. It is possible that these biases rapidly acquire large positive values, which would effectively linearize the final RELU, thus easing signal propagation from the input to the output. We are not sure we entirely understand how to interpret your random walk intuition, but we will contact you to discuss further after our submission is de-anonymised.

---

> > ### Public Comment · ~Antoine_Labatie1 · 2019-11-03
> > **Thank you very much for your response**
> >
> > Thank you very much for your response. I look forward to discussing these ideas further.

---

### Decision · Program_Chairs · 2019-12-19

**Decision:**

Reject

**Comment:**

The paper is rejected based on unanimous reviews.